# Antifibrotic Soluble Thy-1 Correlates with Renal Dysfunction in Chronic Kidney Disease

**DOI:** 10.3390/ijms24031896

**Published:** 2023-01-18

**Authors:** Anja Saalbach, Ulf Anderegg, Ralph Wendt, Joachim Beige, Anette Bachmann, Nora Klöting, Matthias Blüher, Ming-Zhi Zhang, Raymond C. Harris, Michael Stumvoll, Anke Tönjes, Thomas Ebert

**Affiliations:** 1Department of Dermatology, Venereology and Allergology, University of Leipzig Medical Center, 04103 Leipzig, Germany; 2Hospital St. Georg, Division of Nephrology and Kuratorium for Dialysis and Transplantation, 04129 Leipzig, Germany; 3Department for Internal Medicine, Medical Clinic 2, Martin-Luther-University Halle/Wittenberg, 06108 Halle, Germany; 4Medical Department III—Endocrinology, Nephrology, Rheumatology, University of Leipzig Medical Center, 04103 Leipzig, Germany; 5Helmholtz Institute for Metabolic, Obesity and Vascular Research (HI-MAG) of the Helmholtz Zentrum München at the University of Leipzig and University Hospital Leipzig, 04103 Leipzig, Germany; 6Division of Nephrology, Department of Medicine, Vanderbilt University School of Medicine, Nashville, TN 37212, USA; 7Department of Medicine, Nashville Veterans Affairs Hospital, Vanderbilt University School of Medicine, Nashville, TN 37212, USA

**Keywords:** Thy-1, chronic kidney disease, renal dysfunction, fibrosis

## Abstract

Kidney fibrosis is a major culprit in the development and progression of chronic kidney disease (CKD), ultimately leading to the irreversible loss of organ function. Thymocyte differentiation antigen-1 (Thy-1) controls many core functions of fibroblasts relevant to fibrogenesis but is also found in a soluble form (sThy-1) in serum and urine. We investigated the association of sThy-1 with clinical parameters in patients with CKD receiving hemodialysis treatment compared to individuals with a preserved renal function. Furthermore, Thy-1 tissue expression was detected in a mouse model of diabetic CKD (eNOS^−/−^; db/db) and non-diabetic control mice (eNOS^−/−^). Serum and urinary sThy-1 concentrations significantly increased with deteriorating renal function, independent of the presence of diabetes. Serum creatinine is the major, independent, and inverse predictor of serum sThy-1 levels. Moreover, sThy-1 is not only predicted by markers of renal function but is also itself an independent and strong predictor of markers of renal function, i.e., serum creatinine. Mice with severe diabetic CKD show increased *Thy*-1 mRNA and protein expression in the kidney compared to control animals, as well as elevated urinary sThy-1 levels. Pro-fibrotic mediators, such as interleukin (IL)-4, IL-13, IL-6 and transforming growth factor β, increase *Thy*-1 gene expression and release of sThy-1 from fibroblasts. Our data underline the role of Thy-1 in the control of kidney fibrosis in CKD and raise the opportunity that Thy-1 may function as a renal antifibrotic factor.

## 1. Introduction

Kidney fibrosis is a crucial pathomechanism involved in the development and progression of chronic kidney disease (CKD), a global public health problem affecting at least 9.1% of the worldwide population [1]. CKD is associated with a premature vascular aging phenotype, contributing to a substantially higher cardiovascular morbidity and mortality risk in affected individuals [2,3,4,5]. Irrespective of the etiology of CKD, persistent inflammation, hypoxia, and oxidative stress are the drivers for the initiation of kidney fibrosis affecting all compartments of the kidney, leading to arteriosclerosis and glomerulosclerosis and, finally, to the irreversible loss of kidney function [5,6,7]. Pathophysiologically, kidney fibrosis is characterized by an excessive accumulation of stiff, fibrillar extracellular matrix (ECM) and organ dysfunction, contributing to organ failure and death [8,9]. Independent of the trigger, the final common pathways involve the development of myofibroblasts characterized by metabolic alterations, the expression of a contractile machinery, excessive ECM production, and resistance to apoptosis [10]. Understanding the regulatory paradigms controlling the function, as well as the appearance and resolution of fibroblast phenotypes, is important to intervene in this detrimental process.

Thymocyte differentiation antigen-1 (Thy-1, CD90) is a small highly glycosylated protein attached to the outer leaflet of the plasma membrane through a glycosylphosphatidylinositol anchor [11]. It is expressed on fibroblasts, neurons, activated microvascular endothelial cells, glomeruli cells, a subpopulation of hematopoietic stem cells, mesenchymal stem cells, and mouse T-cells. Membrane receptors for Thy-1, such as β2, β3, β5 integrins, CD97, and syndecan-4, have been described. In addition, Thy-1 interacts with molecules within the membrane of the same cell [11]. Functional studies have shown that Thy-1 controls many core functions of fibroblasts relevant to fibrogenesis, including ECM deposition, proliferation, apoptosis, cytokine and growth factor expression and responsiveness, cell adhesion, migration, and myofibroblast differentiation [10]. In vitro, the loss of Thy-1 on lung fibroblasts induces a pro-fibrogenic phenotype including increased proliferation, attenuated apoptosis, increased susceptibility to the pro-fibrotic transforming growth factor (TGF) β, and enhanced myofibroblast differentiation [12,13]. Consistently, Thy-1-deficient mice have developed a more severe lung fibrosis upon intratracheal bleomycin administration than wild-type controls, while the overexpression of Thy-1 attenuated fibrosis in a model of acute interstitial pneumonia [14,15]. Importantly, in human idiopathic pulmonary fibrosis, activated fibroblasts within the fibroblastic foci lose Thy-1 expression [14]. In an experimental model for cardiac hypertrophy and heart failure, Thy-1-negative ventricular fibroblasts accumulated and displayed a more activated, pro-fibrotic myofibroblast phenotype compared to Thy-1-positive cells. Consistently, Thy1-deficient mice developed a more severe cardiac dysfunction and fibrosis compared to WT mice. The loss of Thy-1 expression on cardiac fibroblasts was associated with increased expression of pro-fibrotic genes [16]. Furthermore, Thy-1-deficient mice showed increased liver fibrosis and an upregulation of the expression of profibrogenic genes such as collagen, alpha smooth muscle cell actin (aSMA), mesothelin, and TGF-receptor I.

Regarding kidney fibrosis, Thy-1 expression was described on glomerular mesangial cells [17]. Single-cell RNA sequencing data of healthy kidneys further showed expression in proximal tubular cells and fibroblasts [18]. In a mouse model of unilateral ureteral obstruction-induced kidney fibrosis, a lack of Thy-1 facilitated fibrosis characterized by increased collagen I, aSMA, and TGFβ1 expression [19]. Altogether, the data show that Thy-1 displays anti-fibrogenic properties on fibroblasts.

In addition to the membrane bound form, Thy-1 also exists in a soluble form [20]. Previously, we detected sThy-1 in serum, wound fluid from venous leg ulcers, and synovial fluid from joints in rheumatoid arthritis [21]. Interestingly, sThy-1 concentration is increased in serum of patients with systemic sclerosis compared to healthy controls [21].

Since Thy-1 expression is causally associated with fibrosis development in distinct disease states, we investigated whether tissue Thy-1 expression, as well as serum, urinary, and dialysate sThy-1 mirrors renal dysfunction in a mouse model and in patients with CKD.

## 2. Results

### 2.1. Renal Function Is the Strongest, Inverse Predictor of Circulating sThy-1 Levels in Patients with CKD—Cohort 1 (N = 120)

Serum sThy-1 was analyzed in non-diabetic and diabetic individuals with preserved renal function, as well as with kidney failure on chronic hemodialysis (HD) treatment. Baseline characteristics of the study population of cohort 1, stratified in four subgroups (i.e., non-diabetic healthy controls, patients with type 2 diabetes (T2D) and sustained renal function, as well as non-diabetic and T2D patients with kidney failure on chronic HD treatment), are shown in Table 1. Median (interquartile range) serum levels of sThy-1 in the entire cohort (N = 120) were 9.4 (11.4) µg/L and did not depend on sex (*p* = 0.773). In contrast, patients with CKD on hemodialysis treatment had significantly higher sThy-1 concentrations (16.3 (3.2) µg/L) compared to individuals with an eGFR > 50 mL/min/1.73 m² (5.0 (1.4) µg/mL) (*p* < 0.001). Across the four subgroups in cohort 1, sThy-1 was significantly different (overall *p* < 0.001) and did not depend on T2D status in healthy controls nor in patients on HD treatment (all *p* > 0.05).

To investigate the associations of sThy-1 with anthropometric and biochemical parameters of glucose metabolism, dyslipidemia, inflammation, and renal function, we performed univariate correlations and multivariate linear regression analyses. In univariate correlation analysis, serum levels of sThy-1 were significantly and positively correlated with waist-to-hip ratio (WHR), triglycerides, creatinine, and hsIL-6. In contrast, fasting glucose, total, high-density lipoprotein (HDL), as well as low-density lipoprotein (LDL) cholesterol, and eGFR were negatively related to sThy-1 in univariate analyses, respectively (Table 2, all *p* < 0.05).

To identify independent predictors of sThy-1 in patients with CKD, multiple linear regression analysis was carried out. Here, creatinine was the strongest, positive, and independent predictor of sThy-1 serum levels (*p*<0.001) after adjustment for sex, as well as for markers of obesity, glucose homeostasis, dyslipidemia, and inflammation (Table 2). In addition to creatinine, fasting glucose and HDL cholesterol were further independent, negative predictors, whereas female sex, WHR, and hsIL-6 were positively associated with sThy-1 (Table 2). When eGFR instead of serum creatinine was included in the multivariate model, eGFR as a marker of renal function was again the strongest, negative, and independent predictor of circulating sThy-1 concentrations (standardized β: −0.656; *p* < 0.001).

### 2.2. Urinary sThy-1/Creatinine Levels and Dialysate sThy-1 in Patients with CKD—Cohort 2 (N = 50) and Cohort 3 (N = 6)

As proteinuria is an established marker of an adverse outcome in patients with CKD [5], we next sought to investigate whether sThy-1 can be detected in the urine, and whether urinary sThy-1/creatinine-ratios are related to albuminuria as assessed by albumin-creatinine ratios (ACR) in patients with different stages of CKD. Therefore, the urinary sThy-1/creatinine ratio was quantified in cohort 2, including 10 age-, sex-, and body mass index (BMI)-matched subjects stratified into eGFR categories from G1 to G5. Urinary sThy-1 was detectable in all study participants, and urinary sThy-1/creatinine ratios significantly increased with deteriorating renal function in patients with eGFR category G3–5 compared to eGFR category G1, respectively (Figure 1A). Whereas the overall pattern of sThy-1/creatinine ratios throughout the eGFR spectrum (Figure 1A) was comparable to that of ACR (Figure 1B), urinary ACR was not significantly increased until eGFR category G4 and G5 compared to eGFR category G1, respectively (Figure 1B). Thus, in our cohort, urinary sThy-1/creatinine ratios increased even before urinary ACR was significantly enhanced.

In cohort 3, we analyzed sThy-1 in serum and dialysate. In comparison to paired serum sThy-1 samples, sThy-1 is only marginally detectable in the dialysate about 10 min after start of the hemodialysis session, indicating a low sThy-1 clearance during dialysis (Figure 1C).

### 2.3. Thy-1 Expression Is Increased in a Mouse Model of CKD

To analyze tissue Thy-1 expression during development of diabetic nephropathy, we used the eNOS^−/−^;db/db mouse, a robust model of diabetic nephropathy (CKD) with many similarities to human diabetic kidney disease, including albuminuria, decreased glomerular filtration rate, mesangial expansion, glomerular basement membrane thickening, arteriolar hyalinosis, glomerulosclerosis, and tubulointerstitial injury [22]. In 24-week-old mice, mRNA expression of *collagen type I*(*α*1) and *TGFβ*1 significantly increased in mice with CKD compared to non-diabetic control mice (eNOS^−/−^; Figure 2A). Histologically, kidney fibrosis was confirmed by increased picrosirius-red staining in kidney tissue from eNOS^−/−^; db/db compared to control animals (Figure 2B). In accordance with the histological and gene expression analyses, albuminuria as assessed by ACR was strongly increased in mice with CKD compared to non-CKD controls (Figure 2C).

Thy-1 gene expression in the kidney was significantly increased in mice with CKD compared to littermate controls (Figure 2D). Immunofluorescence staining of the kidney tissue section proved the increased Thy-1 expression in CKD mice (Figure 2E). Immunohistochemical staining of Thy-1 showed strong perivascular (Appendix A) and glomerular expression (Appendix A). Double staining indicates Thy-1 expression in epithelial cells identified by E-cadherin (Appendix A). Moreover, co-localization of Thy-1 with Fibroblast-Specific-Protein (FSP, Appendix A) and collagen III (Appendix A) indicate that Thy-1 was also expressed in fibroblasts.

In addition, urinary sThy-1/creatinine levels were significantly higher in CKD mice compared to eNOS^−/−^ control animals (Figure 2F). In other insulin-sensitive, as well as fibrosis-prone organs, such as the liver, heart, and visceral adipose tissue, Thy-1 mRNA expression was not affected in this model of diabetic nephropathy (Figure 2G).

### 2.4. Regulation of Thy-1 Expression and Release of sThy-1

Our mouse data indicate that sThy-1 expression is increased in CKD mice; thus, sThy-1 might be also a predictor of CKD. Therefore, we investigated whether sThy-1 is a predictor of creatinine using both a model with comparable covariates to our main analysis, as well as a model with well-established covariates predicting CKD identified in the Hordaland Health Study, a European-based cross-sectional cohort of more than 15,000 individuals [23]. In this Danish study, the use of antihypertensive drugs, smoking status, and serum triglycerides were among the most important predictors for serum creatinine in sex-stratified analyses with adjustment for body composition. Thus, we built a second model comprising these variables together with age and sex. As shown in Appendix A, serum sThy-1 is an independent and positive predictor of serum creatinine after adjustment for age, sex, markers of body composition, dyslipidemia, smoking status, and antihypertensive treatment.

Finally, we investigated the regulation of Thy-1 expression and the release of sThy-1 in vitro. Since immunofluorescence staining suggests an increase of Thy-1 expression in fibroblasts, human fibroblasts were stimulated with cytokines relevant in the development of fibrosis and inflammation. As shown in Figure 3, IL-4, IL-13, IL-6, and TGFβ1 stimulated Thy-1 gene expression (Figure 3A) and the release of sThy-1 (Figure 3B). In contrast, pro-inflammatory stimulation with tumor necrosis factor (TNF)α/IL-1β did not affect both (Figure 3A,B).

## 3. Discussion

CKD is characterized by renal damage and an irreversible loss of kidney function. Renal injury initiates an inflammatory cascade, triggering an immune response that results in the production of profibrotic mediators, such as TGFβ by inflammatory cells, resulting in glomerulosclerosis, tubular atrophy, and an irreversible state of interstitial fibrosis [9,24].

Thy-1 is a membrane protein controlling the differentiation, apoptosis, and proliferation of fibroblasts, the production of ECM and the responsiveness to fibrogenic mediators [12,25,26,27,28]. All of these functions are central in the development of fibrosis. In addition, a soluble form of Thy-1 has been detected in serum, in urine, as well as in wound, synovial, and cerebrospinal fluids [21,29,30]. In the present study, we showed that serum sThy-1 was significantly elevated in patients with kidney failure on hemodialysis treatment. Importantly, circulating sThy-1 was increased in both diabetic and non-diabetic patients with CKD. In accordance with our data, Wu et al. showed that plasma levels of sThy-1 were significantly higher in patients with diabetic kidney disease compared with diabetic patients without impaired renal function and healthy controls [18]. In our cohort, multiple linear regression analysis identified serum creatinine as the strongest, positive, and independent predictor of sThy-1 serum levels independent of sex, as well as markers of obesity, glucose homeostasis, dyslipidemia, and inflammation. In contrast, in the cohort of Wu et al. involving only patients with diabetes, no correlation between plasma sThy-1 and clinical parameters of diabetic kidney disease was found. The differences might be explained by the study population due to the severity of the disease [18]. However, Wu et al. found a strong correlation of urinary sThy-1 with parameters of impaired renal function, such as eGFR, ACR, and serum creatinine [18]. Consistently, we demonstrated a significant increase of urinary sThy-1/creatinine ratios with deteriorating renal function. Importantly, urinary sThy-1 started to increase as early as eGFR category G3. These findings indicate that serum and urinary sThy-1 are closely related to the development and progression of CKD. It is interesting to note in this context that not only creatinine serum levels were independently associated with sThy-1 in our cohort but circulating sThy-1 was also an independent and positive predictor of serum creatinine after adjustment for age, sex, markers of body composition, dyslipidemia, smoking status, and antihypertensive treatment. This further supports our hypothesis of a closed and bidirectional link of renal function with serum sThy-1 levels.

Different mechanisms may be relevant for the observed increase of sThy-1 in CKD. In our animal model of diabetic nephropathy, we observed both a strong increase of Thy-1-positive cells and Thy-1 gene expression. Double stainings indicate an expression in epithelial cells and fibroblasts. Consistently, an elevated expression of Thy-1 has been reported in several mouse models of fibrosis, such as Mdr2 deficiency-induced liver fibrosis, bleomycin-induced lung fibrosis, and unilateral urinary obstruction-induced kidney fibrosis [19]. Consistently, in vitro data show an upregulation of Thy-1 gene expression and sThy-1 release in fibroblasts by pro-fibrotic mediators such as IL-4, IL-13, and TGFβ. Indeed, we found elevated expression of TGFβ1 in our CKD mouse model. IL-4 and IL-13 were not detectable (data not shown). In addition, IL-6 stimulated Thy-1 gene expression and release. The correlation of sThy-1 and IL-6 in serum of our patients underlines the role of IL-6 in the control of Thy-1 expression.

Taking into consideration that Thy-1-deficient mice develop more severe kidney fibrosis, it is tempting to speculate that renal Thy-1 upregulation in CKD is a compensatory mechanism to limit the adverse consequences of the pro-fibrotic status in CKD. Several mechanisms have been already described. The control of apoptosis by Thy-1 in lung and dermal fibroblasts via Fas-, Bcl-, and caspase-dependent pathways has been demonstrated [31]. Consistently, Thy-1-deficient mice displayed increased fibroblast proliferation, decreased apoptosis, and downregulation of FasL [25,32]. The reduced apoptosis of Thy-1-negative myofibroblasts resulted in the persistence of myofibroblasts during the resolution phase of bleomycin-induced lung fibrosis, was associated with collagen accumulation, and extended lung fibrosis in Thy-1-deficient mice [31,33]. Moreover, the interaction of Thy-1 *in cis* controls fibrogenic actions of fibroblasts. In detail, the disruption of Thy-1–αvβ3-integrin-interaction is sufficient to induce myofibroblast differentiation in soft ECMs [34]. The interaction of Thy-1 with integrin αvβ5 *in cis* inhibits fibroblast contraction-induced latent TGFβ1 activation and TGFβ1-dependent lung myofibroblast differentiation [27]. Moreover, Thy-1-negative but not Thy-1-positive rat lung fibroblasts respond to profibrotic cytokines with the activation of latent TGFβ1 and exhibit higher myofibroblast gene expression and higher collagen contraction than Thy-1-positive cells [12,28]. Koyama et al. demonstrate that under physiological conditions Thy-1 forms an inhibitory complex with TGFβRI [35]. TGFβ1 stimulation facilitates the binding between mesothelin (Msln) and Thy-1, thereby promoting dissociation of Thy-1 from TGFβRI and enabling TGFβ1/TGFβRI signaling in lung and portal fibroblasts. Deletion of Msln in activated portal fibroblasts results in the strong overexpression of Thy-1, which in turn binds to TGFβRI and prevents TGF-β1 signaling [19,35]. In our animal model of diabetic nephropathy, we observed both a strong increase of Thy-1-positive cells and Thy-1 gene expression as observed in the unilateral urinary obstruction-induced (UUO) kidney fibrosis. Deletion of Thy-1 strengthens fibrosis, while the lack of Msln reduces fibrosis in this model. Moreover, the deletion of both resulted in a phenotype similar to the wild-type animals, suggesting that Msln and Thy-1 mediate opposing functions in kidney fibroblasts [19]. We suppose that similar mechanism might act in our mouse model of CKD. In summary, Thy-1 restricts myofibroblast differentiation and favors the resolution of fibrosis.

Thy-1 is a GPI-anchored protein, which can be cleaved by phospholipase C or D [36]. The cleavage of the GPI anchor induces large conformational changes [37] and thus, the function of the sThy-1 might be different from the membrane bound Thy-1. Increase of Thy-1 expression during the development of fibrosis or alteration of shedding activity might contribute to enhanced levels of circulating sThy-1. The increase of urinary sThy-1 might be due to the disturbed glomerular barrier function in CKD patients. Due to its reported molecular weight (i.e., 25–37 kDa), renal elimination of sThy-1 is plausible [38]. Indeed, low concentrations of sThy-1 in dialysate during hemodialysis treatment suggest that sThy-1 is dialyzable. Therefore, future pathophysiological studies need to determine the elimination of sThy-1. It should be noted, however, that single cell sequencing data and immunofluorescence stainings show Thy-1 expression in tubular cells in addition to fibroblasts [18]. However, the role of proximal tubular cells in the generation of sThy-1 is still unknown.

In conclusion, circulating and urinary sThy-1 are significantly associated with impaired renal function, and renal expression of Thy-1 is increased in kidney fibrosis. These data raise the opportunity that Thy-1 may function as a renal-protective factor against fibrosis. Based on our data, future studies on sThy-1 need to adjust data for renal function as a major confounding factor.

## 4. Material and Methods

### 4.1. Human Studies

#### 4.1.1. Cohort 1

For the cross-sectional analysis of patients with kidney failure on hemodialysis (HD) treatment compared to control individuals, about 120 patients were recruited by the Department of Endocrinology and Nephrology, University of Leipzig as part of a larger cohort [39]. The study design of this cohort has been described previously [40,41,42]. Briefly, inclusion criteria were an age > 18 years, nonpregnant, and written informed consent. Patients with end-stage malignant diseases, acute generalized inflammation, acute infectious disease, and a history of drug abuse were excluded from the study. In all individuals, blood specimens were taken after an overnight fast. In all hemodialysis patients, blood was obtained just before hemodialysis started. The study was approved by the local ethics committee of the University of Leipzig (Reg. No: 180-13-15072013, also valid for cohort 2), and all subjects gave the written informed consent before taking part in the study.

#### 4.1.2. Cohort 2

In a sub-cohort of a cross-sectional study [39,43], patients were classified based on their eGFR into eGFR categories G1–G5 according to the Kidney Disease Improving Global Outcomes guidelines [44]. Spot urine samples were obtained in 10 age-, sex-, and BMI-matched subjects per eGFR category.

#### 4.1.3. Cohort 3

To investigate the potential mechanisms of sThy-1 regulation in patients with kidney failure, six patients on chronic hemodialysis were recruited from the KfH renal center at St. Georg Hospital Leipzig, Leipzig, Germany. Patients were included if they were under stable conditions on thrice-weekly hemodialysis without signs of infection or acute diseases. Hemodialysis was performed with Nikkiso DBB-05 and DBB-EXA dialysis machines (Nikkiso Europe, Langenhagen, Germany) using a high-flux filter (Nipro Elisio 19H, Nipro, Mechelen, Belgium). All patients fasted for at least 6 h before blood and dialysate collection. Paired blood and dialysate samples were collected within 10 to 20 min after the start of each hemodialysis session. The study was approved by the Ethical Committee of the Saxonian Board of Physicians (EK-BR-14/20-3), and all subjects gave written informed consent before taking part.

Routine serum parameters, including creatinine, fasting glucose, triglycerides, total, HDL, and LDL cholesterol, as well as urinary albumin and creatinine, were measured in a certified laboratory by standard methods. High-sensitivity IL-6 was determined using enzyme-linked immunosorbent assays according to the manufacturer’s instructions (hsIL-6, R&D Systems, Minneapolis, MN, USA).

### 4.2. Detection of Human sThy-1

High-binding ELISA plates (Sarstedt, Nürnbrecht, Germany) were coated with 0.25 μg/well anti-Thy-1 antibody in 0.1 M NaHPO_4_/0.1 M NaH_2_PO_4_ pH 9.0 (clone AS02; Dianova, Hamburg, Germany) overnight at 4 °C. Plates were washed three times with PBS/0.05% Tween 20 and blocked with PBS/10% fetal calf serum (FCS) for 1 h at 4 °C. After several washes with PBS/0.05% Tween 20, samples and standard (recombinant Thy-1-Fc (R&D, Minneapolis, MN, USA, 5000 pg/mL − 7.8 pg/mL) were diluted in PBS/10 % FCS and incubated overnight at 4 °C. The plates were washed five times with PBS/0.05% Tween 20. Then, 0.5 µg/mL biotinylated anti-CD90 monoclonal antibody (clone 5E10, Pharmingen, Hamburg, Germany) was added for 90 min at room temperature. After three washes with PBS/0.05% Tween 20, avidin-conjugated peroxidase (Invitrogen, Carlsbad, CA, USA) 1:500 diluted in PBS/10 % FCS was added for 1 h at room temperature. Plates were rinsed five times with PBS/ 0.05% Tween 20. Subsequently, tetramethylbenzidine was used to generate the color reaction. After 10 min, the reaction was stopped with 1 M H_2_SO_4,_ and samples were measured at 405 nm.

### 4.3. Animal Studies

For this study, eNOS^−/−^; db/db mice on a C57BLKS background served as a model for severe CKD due to diabetic nephropathy [22,45], whereas non-diabetic eNOS^−/−^ mice were used as control animals. All animal experiments were performed in the Medical Experimental Center at the University of Leipzig and approved by the local ethics committee of the state of Saxony (Landesdirektion Leipzig; approval no. TVV 12/14 and TVV 30/19). At the age of 12 and 24 weeks, spot urine samples were obtained in the morning. At 24 weeks, blood was collected under deep anesthesia. Liver, kidney, heart, visceral, and adipose tissue were snap-frozen in liquid nitrogen for later RNA isolation. In addition, the kidney was frozen for the preparation of cryosections. Urinary albumin and creatinine were determined in spot urine using Albuwell M and Creatinine Companion kits (Exocell, Philadelphia, PA, USA) and albumin-creatinine ratios (ACR) were calculated, respectively.

### 4.4. Staining of Tissue Sections

Cryostat sections of kidney were fixed with acetone and incubated with anti-Thy-1 antibody-PE (BD). For double staining anti-Thy-1 (Pharmingen), anti-fibroblast-specific protein (Millipore, Darmstadt, Germany) and anti-collagen III (Abcam, Cambridge, UK) antibodies were used followed by goat-anti-rat Alexa 488 or goat-anti-rabbit Alexa 488 (Invitrogen), respectively. Subsequently, CD324-PE (BD) or anti-Thy-1-PE (BD) was added. Nuclei were stained with DAPI (Merck, Berlin, Germany). Microscopy was performed with Keyence BZ-9000E and corresponding software BZ-II Viewer, BZ-II Analyser (Keyence, Leipzig, Germany). Collagen deposition was detected by Picro Sirius Red staining (Morphisto, Offenbach, Germany).

### 4.5. Gene Expression

Total RNA of mouse tissue was isolated with TRIzol reagent (Invitrogen, Life Technologies, Carlsbad, CA, USA). Total RNA was further purified by the RNeasy Mini Kit (Qiagen, Hilden, Germany), and 1.5 µg RNA was reverse transcribed using standard reagents (Invitrogen, Life Technologies). The following primer pairs were used: *Thy-1* (Forward primer: TCCATCCAGCATGAGTTC; Reverse primer: TGAACCAGCAGGCTTATG); TGFβ1 (Forward primer: AAGTTGGCATGGTAGCCCTT; Reverse primer: GCCCTGGATACCAACTATTGC), *collagen type* I(α1) (Forward primer: GAGCGGAGAGTACTGGATCG; Reverse primer: GTTCGGGCTGATGTACCAGT); *36B4* (Forward primer: AAGCGCGTCCTGGCATTGTCT; Reverse primer: CCGCAGGGGCAGCAGTGGT). Gene expression in mouse tissue was detected relative to acidic ribosomal phosphoprotein P0 (*36B4*) using quantitative real-time RT-PCR and a fluorescent temperature cycler (Roche, Heidelberg, Germany).

RNA from cultured cells was isolated using the ReliaPrep RNA Miniprep system (Promega, Walldorf, Germany) according to the manufacturer’s protocol. In addition, 0.5 µg RNA was used for cDNA synthesis using LunaScript RT Supermix (New England Biolabs, Frankfurt am Main, Germany) according to the manufacturer’s instructions. Real-time qPCR was performed with LunaUniversal qPCR Mastermix (New England Biolabs, Ipswich, MA, USA) following the company’s instructions. Quantitative gene expression was calculated from standard curve of cloned cDNA and normalization to the reference gene RPLP0. The following primers were used: human *Thy-1* (Forward primer: ATGAACCTGGCCATCAGCATCGC; Reverse primer: CGAGGTGTTCTGAGCCAGCAGGC), human *rps26* (Forward primer: CAATGGTCGTGCCAAAAAG; Reverse primer: TTCACATACAGCTTGGGAAGC).

### 4.6. Cell Culture

Human skin biopsies were incubated with 2.2 U/mL dispase II (Roche) overnight at 4 °C. After removal of the epidermal layer, the dermis was digested with 2.5 mg/mL collagenase (Sigma, Darmstadt, Germany) for 45 min at 37 °C. Cell suspension was passed through a 70 µm filter. Cells were cultured at 37 °C, 5% CO_2_ in DMEM medium (Anprotec, Bruckberg, Germany) containing 10% fetal calf serum (FCS; PAN-Biotech, Aidenbach, Germany) and 1% penicillin/streptomycin (Biochrom, Berlin, Germany). After reaching confluence, cells were passaged using 0.05% trypsin and 0.02% EDTA (Biochrom). Cells were stimulated with 10 ng/mL human IL-4 (Miltenyi, Bergisch-Gladbach, Germany), IL-6 (Miltenyi), IL-13 (Miltenyi), or TNFα/IL1β (Miltenyi) or 20 ng/mL TGFβ1 (Peprotech, Hamburg, Germany) in DMEM/0.5 % FCS for 9d. Medium was changed after 7d. To exclude any effects of proliferation or cell death, the cell number was counted, and the sThy-1 concentration was presented as pg per 10,000 cells.

### 4.7. Statistical Analyses

IBM SPSS Statistics software version 28.0 (IBM, Armonk, NY, USA) was used for all statistical analyses in human subjects, and the tests used are described in the respective table and figure legends and in previous studies from our group [39,41,42,43]. GraphPad Prism 9 (GraphPad Software, San Diego, CA, USA) was used for all statistical analyses in mice and in vitro experiments, and the tests used are described in the respective figure legends. *p*-values < 0.05 were considered statistically significant in all analyses.

## Figures and Tables

**Figure 1 ijms-24-01896-f001:**
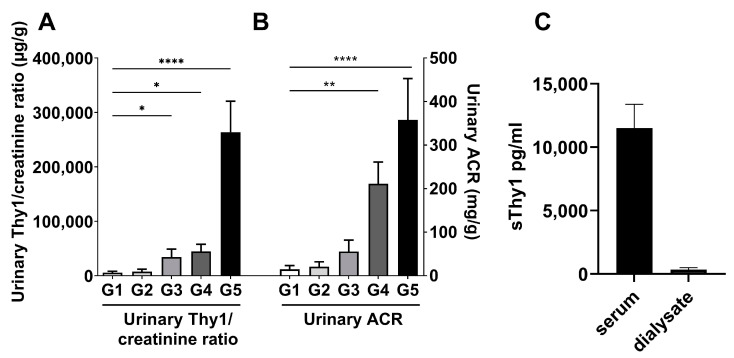
Urinary sThy-1 and dialysate sThy-1 in patients with CKD. Urinary losses of sThy-1 and albumin were detected in spot urine samples of patients with estimated glomerular filtration rate (eGFR) categories G1–G5 (N = 10/eGFR category, all groups G1–G5 were age-, sex- and body mass index-matched). (**A**) Urinary sThy-1/creatinine and (**B**) urinary albumin to creatinine ratio (ACR) were determined in cohort 2 (N = 50). (**C**) The sThy-1 in paired blood and outflow dialysate sampling 10–20 min after start of hemodialysis in patients with kidney failure receiving hemodialysis treatment were analyzed in cohort 3 (N = 6). All results are shown as mean ± standard deviation. One-way ANOVA test in (**A**,**B**). * *p* < 0.05; ** *p* < 0.01; **** *p* < 0.0001.

**Figure 2 ijms-24-01896-f002:**
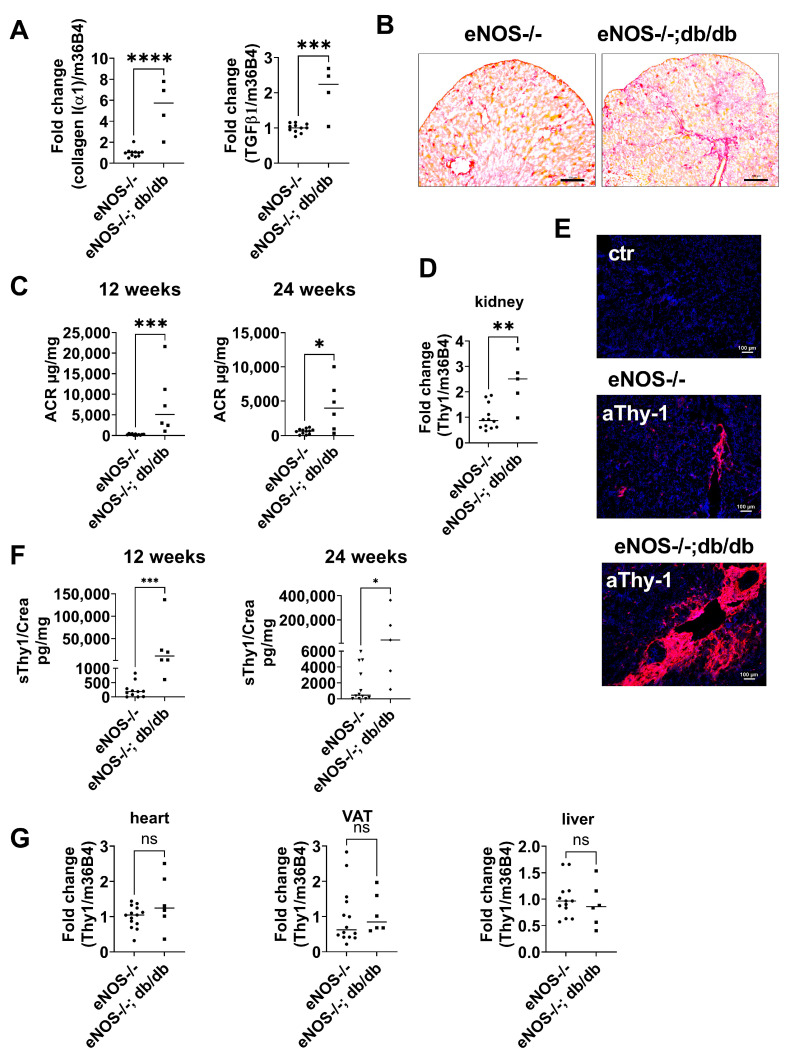
Thy-1 is upregulated in a mouse model of chronic kidney disease (CKD) due to diabetic nephropathy compared with non-diabetic control mice. (**A**) Relative *collagen type I*(*α*1) and *TGFβ*1 gene expression in the kidney of CKD (eNOS^−/−^ C57BLKS db/db mice) and control mice (eNOS^−/−^ C57BLKS) detected by qPCR at the age of 24 weeks. (**B**) Detection of collagen by Picro Sirius red staining in tissue sections of CKD and control mice. One representative example of four mice is shown. bar = 500 µm. (**C**) Urinary albumin to creatinine ratio (ACR) in spot urine samples of 12- and 24-week-old CKD and control mice. (**D**) Relative *Thy*-1 gene expression in kidney detected by qPCR. (**E**) Thy-1 protein expression in kidney detected by immunofluorescence labelling with anti-Thy-1 antibody (red). Nuclei (blue) were counterstained with DAPI. Images represent one representative example of four different mice. (**F**) Urinary sThy-1/creatinine ratio in spot urine samples of CKD and control mice detected at the age of 12 and 24 weeks by ELISA. (**G**) Relative *Thy*-1 gene expression in heart, liver, and visceral adipose tissue (VAT) detected by qPCR. Results are shown as mean ± standard deviation. The *p*-values were assessed by unpaired Student’s *t*-tests. Each point represents one mouse. * *p* < 0.05; ** *p* < 0.01; *** *p* < 0.001, **** *p* < 0.0001. bar = 100 µm.

**Figure 3 ijms-24-01896-f003:**
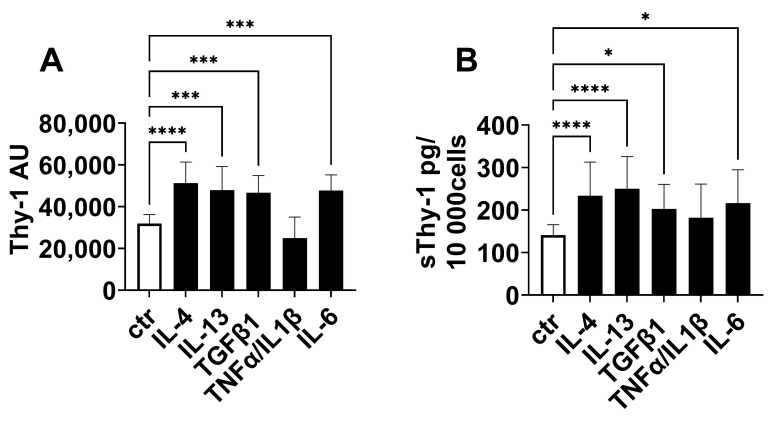
Thy-1 was upregulated in fibroblasts by pro-fibrotic mediators. Human fibroblasts were stimulated with interleukin (IL)-4, IL-13, IL-6, TGFβ1 or tumor necrosis factor (TNF)α/IL1β for 9d. (**A**) Thy-1 gene expression was detected by PCR. (**B**) Soluble Thy-1 (sThy-1) was determined in supernatants by ELISA. The cell number was counted, and sThy-1 is presented as pg per 10,000 cells. Results are shown as mean ± standard deviation. *p*-values were assessed by one-way ANOVA test. N ≥ 5 per group. * *p* < 0.05; *** *p* < 0.001, **** *p* < 0.0001.

**Table 1 ijms-24-01896-t001:** Baseline characteristics of the study population in cohort 1 (N = 120).

	Subgroup 1 Control/nonT2D	Subgroup 2 Control/T2D	Subgroup 3 HD/nonT2D	Subgroup 4 HD/T2D	Overall *p*
N	30	30	28	32	-
Sex (male/female)	11/19	16/14	15/13	20/12	0.239
sThy-1 (µg/L)	5.2 (1.7) ^3,4^	4.6 (1.2) ^3,4^	16.4 (3.3) ^1,2^	16.3 (3.8) ^1,2^	**<0.001**
Age (years)	63 (19)	63 (16)	59 (23)	68 (12)	0.051
BMI (kg/m²)	28.2 (5.6)	29.1 (5.2) ^3^	25.2 (6.5) ^2^	27.9 (6.6)	**0.004**
WHR	0.88 (0.12) ^4^	0.94 (0.10)	0.96 (0.18)	1.00 (0.14) ^1^	**<0.001**
SBP (mmHg)	125 (21)	126 (20)	125 (38)	120 (25)	0.339
DBP (mmHg)	77 (10)	73 (15)	77 (20)	70 (18)	0.095
FG (mmol/L)	5.1 (1.3) ^2^	7.6 (3.2) ^1,3,4^	4.6 (1.2) ^2^	5.2 (3.3) ^2^	**<0.001**
Total cholesterol (mmol/L)	5.3 (0.9) ^3,4^	4.9 (1.5)	4.4 (1.1) ^1^	4.2 (1.3) ^1^	**<0.001**
HDL cholesterol (mmol/L)	1.4 (0.4) ^3,4^	1.2 (0.5) ^4^	1.0 (0.5) ^1^	1.0 (0.3) ^1,2^	**<0.001**
LDL cholesterol (mmol/L)	3.5 (1.1) ^2,3,4^	2.9 (0.9) ^1^	2.7 (0.9) ^1^	2.1 (1.4) ^1^	**<0.001**
TG (mmol/L)	1.1 (0.8) ^3,4^	1.4 (0.9)	1.6 (0.9) ^1^	1.8 (1.4) ^1^	**<0.001**
Creatinine (μmol/L)	76 (17) ^3,4^	72 (22) ^3,4^	829 (431) ^1,2^	717 (221) ^1,2^	**<0.001**
eGFR (ml/min/1.73 m^2^)	78.8 (24.4) ^3,4^	85.2 (23.0) ^3,4^	5.0 (3.2) ^1,2^	5.7 (2.7) ^1,2^	**<0.001**
hsIL-6 (ng/L)	1.87 (2.11) ^3,4^	2.07 (1.25) ^3,4^	6.14 (4.91) ^1,2^	7.48 (7.30) ^1,2^	**<0.001**

Baseline characteristics of the study population in cohort 1 (N = 120) stratified in the four subgroups, i.e., Control/nonT2D, Control/T2D, HD/nonT2D, and HD/T2D. BMI, Body mass index; DBP, Diastolic blood pressure; eGFR, Estimated glomerular filtration rate; FG, Fasting glucose; HD, Hemodialysis; HDL, High density lipoprotein; hsIL-6, High sensitivity interleukin-6; LDL, Low density lipoprotein; SBP, Systolic blood pressure; sThy-1, Soluble Thy-1 cell surface antigen; T2D, Type 2 diabetes; TG, Triglycerides; WHR, waist-to-hip ratio. Values for median (interquartile range) or total numbers are shown. Categorical parameters were analyzed using the χ²-test. Continuous parameters were analyzed by the Kruskal–Wallis test followed by Bonferroni-adjusted post-hoc analysis. Overall *p* values of the Kruskal–Wallis test are depicted. Superscript numbers (^1,2,3,4^) indicate subgroup comparisons with significant Bonferroni-adjusted post-hoc analyses. ^1,2,3,4^ indicate *p* < 0.05 as compared to the ^1^ Subgroup 1 (control/nonT2D), ^2^ Subgroup 2 (control/T2D), ^3^ Subgroup 3 (HD/nonT2D), and ^4^ Subgroup 4 (HD/T2D), respectively.

**Table 2 ijms-24-01896-t002:** Univariate correlations and multivariate linear regression analysis of serum sThy1 with anthropometric parameters and markers of glucose metabolism, serum lipids, inflammation, and renal function in cohort 1 (N = 120).

	Univariate Correlation Analyses		Multivariate Linear Regression Analysis	
	*r*	*p*	**β**	*p*
Age (years)	0.112	0.224	-	-
Sex	-	-	**0.143**	**<0.001**
BMI (kg/m^2^)	−0.142	0.122	-	-
WHR	**0.208**	**0.022**	**0.065**	**0.01**
Fasting glucose (mmol/L)	**−0.352**	**<0.001**	**−0.157**	**<0.001**
SBP (mmHg)	−0.081	0.377	-	-
DBP (mmHg)	−0.151	0.1	-	-
Total cholesterol (mmol/L)	**−0.26**	**0.004**	-	-
HDL cholesterol (mmol/L)	**−0.374**	**<0.001**	**−0.161**	**<0.001**
LDL cholesterol (mmol/L)	**−0.292**	**0.001**	-	-
TG (mmol/L)	**0.297**	**<0.001**	-	-
Creatinine (µmol/L)	**0.777**	**<0.001**	**0.662**	**<0.001**
eGFR (ml/min/1.73 m²)	**−0.81**	**<0.001**	-	-
hsIL-6 (ng/L)	**0.752**	**<0.001**	**0.329**	**<0.001**

Univariate correlations and multivariate linear regression analysis of sThy1 with anthropometric and biochemical markers in cohort 1 (N = 120). Non-parametric Spearman’s rank correlation method was used to assess univariate relationships between sThy-1 and indicated markers. Multivariate regression analysis was calculated for sThy-1 (lg, dependent variable) adjusted for sex, WHR (lg), fasting glucose (lg), HDL cholesterol (lg), creatinine (lg), as well as hsIL-6 (lg). Non-normally distributed variables as assessed by the Shapiro–Wilk test were logarithmically transformed prior to multivariate testing (lg). The *r*- and *p*-values, as well as standardized β-coefficients and *p*-values, are given. The coefficients corresponding to sex assume that males are coded such that females have a larger value. Abbreviations are indicated in Table 1.

## Data Availability

Data is contained within the article and Appendix A.

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
