# Peer review of "Antifibrotic Soluble Thy-1 Correlates with Renal Dysfunction in Chronic Kidney Disease"

_ijms, 2023, doi:10.3390/ijms24031896_

Round 1
Reviewer 1 Report (Previous Reviewer 2)
No further comment.
Author Response
Thanks for the evaluation of our manuscript.
Attached you find the final version after all changes requested by the reviewers. Reviewers 2 suggested following changes:
- In Figure 2B, the picture of the eNOS-/- group looks not good. I suggest the authors replace it. And In Figures 2B and 2E, the scale bars are unclear; please replace them.
We included information about bar length in the Figure Legend. In addition, we included thicker bars into the images.
We replaced Fig 2B by another image.
- It is better to present the individual data as dot plots (scatter plots).
Thanks for the suggestion. We changed the diagrams in Figure 2 to dot plots
Best regards,
Anja Saalbach
Reviewer 2 Report (New Reviewer)
The authors investigated the association of sThy-1 with clinical parameters in patients with CKD receiving hemodialysis treatment compared to individuals with preserved renal function. Moreover, they found renal expression of Thy-1 is increased in kidney fibrosis.
Furthermore, they demonstrated that sThy-1 concentrations significantly increased with deteriorating renal function, independent of the presence of diabetes. These data showed that Thy-1 might function as a renal-protective factor against fibrosis.
Overall, it is an interesting and clear article.
1. In Figure 2B, the picture of the eNOS-/- group looks not good. I suggest the authors replace it. And In Figures 2B and 2E, the scale bars are unclear; please replace them.
2. It is better to present the individual data as dot plots (scatter plots).
Author Response
Thanks for the positive evaluation of our study.
- In Figure 2B, the picture of the eNOS-/- group looks not good. I suggest the authors replace it. And In Figures 2B and 2E, the scale bars are unclear; please replace them.
We included information about bar length in the Figure Legend. In addition, we included thicker bars into the images.
We replaced Fig 2B by another image.
- It is better to present the individual data as dot plots (scatter plots).
Thanks for the suggestion. We changed the diagrams in Figure 2 to dot plots.
Best regards,
Anja Saalbach
This manuscript is a resubmission of an earlier submission. The following is a list of the peer review reports and author responses from that submission.
Round 1
Reviewer 1 Report
Saalbach et al found soluble Thy-1 increased in patients with type 2 diabetes, and associated with the loss of renal function. Furthermore, they found sThy-1 increased in serum, urine, and diabetic mice. Although this work is potentially interesting, it is not fully supportive for the conclusion. Major concerns:
1. Although serum Thy-1 was correlated with serum creatinine, it did not show a correlation with eGFR.
2. Urinary Thy-1 only increased in CKD stage G5, suggesting it was not a sensitive marker.
3. Thy-1 increased in db/db mice model, however, what is its function? How it affected renal fibrosis?
4. Serum Thy-1 showed a correlation with serum IL-6, an inflammation marker. However, Thy-1 could not be induced under inflammatory condition, such as TNFα/IL1β stimulation. What is the reason?
5. What is the cell origin for Thy-1 expression? The co-localizing should be performed.
6. This work only showed a very preliminary data for Thy-1 detection in serum, urine, diabetic mouse kidneys. However, how Thy-1 affect renal cell function? What are the mechanisms? From this manuscript, the reviewers could not get novel information on the above issues.
Reviewer 2 Report
Anja et al. investigated the relationship between sThy-1 and renal function in CKD patients. They also validated the results in the animal model. The results are solidate and convincing. However, I have some concerns.
1. In the animal model, the found that sThy-1 expression is increased in CKD mice. I believe the purpose of animal experiment is verify the sThy-1 is a predictor or risk factor of CKD. But in the clinical cohort, they analyzed the independent factor of sThy-1. That means sThy-1 is the end point parameter rather than a predictor of CKD. I am very confused.
2. What is the mechanism of sThy-1 in the anti-fibrotic of kidneys? I cannot get any data from the animal study.